# Vertical eddy iron fluxes support primary production in the open Southern Ocean

Takaya Uchida ⬡ [1✉], Dhruv Balwada[2], Ryan P. Abernathey ⬡ [1,3], Galen A. McKinley ⬡ [1,3], Shafer K. Smith[2] & Marina Lévy[4]

The primary productivity of the Southern Ocean ecosystem is limited by iron availability. Away from benthic and aeolian sources, iron reaches phytoplankton primarily when iron-rich subsurface waters enter the euphotic zone. Here, eddy-resolving physical/biogeochemical simulations of a seasonally-forced, open-Southern-Ocean ecosystem reveal that mesoscale and submesoscale isopycnal stirring effects a cross-mixed-layer-base transport of iron that sustains primary productivity. The eddy-driven iron supply and consequently productivity increase with model resolution. We show the eddy flux can be represented by specific well-tuned eddy parametrizations. Since eddy mixing rates are sensitive to wind forcing and large-scale hydrographic changes, these findings suggest a new mechanism for modulating the Southern Ocean biological pump on climate timescales.

[1] Department of Earth and Environmental Sciences, Columbia University in the City of New York, New York City, NY, USA. [2] Center for Atmosphere Ocean Science, Courant Institute of Mathematical Sciences, New York University, New York City, NY, USA. [3] Division of Ocean and Climate Physics, Lamont-Doherty Earth Observatory, Palisades, NY, USA. [4] Laboratoire d'Océanographie et du Climat, Institut Pierre Simon-Laplace, Paris, France. ✉email: takaya@ldeo.columbia.edu

Budgets of iron, the limiting nutrient in the Southern Ocean for primary production[1–3], estimated from shiptrack observations have emphasized the importance of dust deposition, lateral transport, and recycling of iron, concluding that contributions from upwelling are negligible[4]. More recently, however, one-dimensional process studies have highlighted the importance of mixed-layer entrainment[5] and vertical diffusion of iron[6] in regions remote from dust sources. Due to the sparse spatial and temporal coverage of in situ iron observations and the intermittent nature of iron supply and phytoplankton blooms, a basin-scale view has generally relied on global circulation models (GCMs[7,8]) and data assimilation products[9]. A GCM intercomparison study showed that, although the iron sources and biogeochemical parameters varied widely, the global-mean iron concentrations were largely in agreement, a consequence of model tuning towards this target[10]. When compared against individual ocean transects, however, the GCMs showed a large inter-model disagreement. This spread was attributed to differences in how each model represented the scavenging of iron. Due to computational constraints, eddy tracer transport in GCMs must be parametrized, also potentially causing uncertainty in the physical processes transporting iron[11] and resulting ecosystem.

In addition to vertical diapycnal mixing and large-scale circulation, mesoscale eddies (on scales of roughly 20–200 km and to first-order geostrophically balanced) can make a major contribution to tracer transport[12,13]. In the Southern Ocean, upward vertical mesoscale eddy heat fluxes counteract the downward flux of heat due to Ekman pumping[14], and mesoscale eddies help regulate the subduction of anthropogenic carbon from the surface into the interior[11,15]. At even smaller scales where the geostrophic approximation breaks down, submesoscale turbulence (roughly 1–20 km and associated with Rossby and Richardson numbers on the order of unity) generates vigorous vertical velocities near the surface[16,17]. In the North Atlantic, submesoscale turbulence has been argued to drive significant transport of nutrients across the mixed-layer base, supporting ecosystem productivity[18]. Do eddies play the same role with iron in the Southern Ocean?

To our knowledge, this question has only been investigated by examining Lagrangian particle trajectories from a high-resolution numerical simulation of the Kerguelen region. Calculating iron concentration in the reference frame of Lagrangian particles, Rosso et al.[19,20] argued that submesoscale iron fluxes could enhance primary productivity by a factor of two. While suggestive, their simulation technique did not implement a full ecosystem model, account for the strong seasonal cycle in both turbulence and biology, nor include fluxes from vertical mixing or mixed-layer entrainment. The relative contribution of eddies to the open-Southern-Ocean primary productivity therefore warrants further investigation.

Here we take a different approach: we run a state-of-the-art numerical simulation at submesoscale permitting resolution in an idealized channel configuration and force the model with a realistic seasonal cycle. Due to the approximate zonal symmetry of the Antarctic Circumpolar Current, such configurations can capture the broad characteristics of Southern Ocean circulation, tracer transport, and ventilation[21,22]. The reduced computational cost (compared to a global-scale simulation) enables our model to reach physical and biogeochemical equilibrium, and the simple geometry facilitates straightforward interpretation of the dynamics. By varying the model resolution, we resolve, suppress, or parametrize the eddies and show that eddy iron transport modulates primary production in the open Southern Ocean.

## Results

**Submesoscale permitting simulation of the open Southern Ocean ecosystem.** We use the Masachusetts Institute of Technology

general circulation model[23] (MITgcm) with an embedded full ecosystem model[24,25]. The model configuration is identical to a companion paper[26] where we quantify the relative contribution of submesoscale and mesoscale dynamics on the total vertical iron transport. For completeness, details of the set up are also summarized in Supplementary Note 1. In this study, we focus on the biogeochemical effect of eddy iron transport on primary production and whether eddy parametrizations in non-eddying runs can replicate this unresolved flux. A snapshot of the phytoplankton biomass and iron field in the top 300 m on 1 November from the 2 km run, during the height of spring bloom, is shown in Fig. 1. The Rossby deformation radius at the center of the domain is 14 km, so the horizontal resolution of 2 km allows us to observe the imprint of mesoscale coherent features[17], such as fronts and eddies, in both iron and phytoplankton.

To simulate the interaction of this region with the rest of the ocean, iron and other nutrients are relaxed to climatological observational profiles at the Northern boundary; in the rest of the domain their concentrations evolve freely based on the simulated circulation and ecosystem. In order to isolate the role of open-ocean transport processes, we do not supply aeolian dust input at the surface or glacial and bathymetric iron sources from the South. The annual-zonal-mean iron transect (Fig. 1c) shows enhanced iron concentrations at depth and strong depletion near the surface. A comparison with GEOTRACES iron profiles from the Southern Ocean (Fig. 1d) indicates that our simulation has a realistic ferrocline structure, in contrast with most of the global-scale GCM simulations[10]. Deep iron concentrations of roughly 0.4 μmol Fe m$^{-3}$ at 1000 m coincide with the observational mean in the ACC, while near-surface concentrations (0.05 μmol Fe m$^{-3}$) are lower than the observational range. This discrepancy is likely due to the lack of aeolian, glacial, and bathymetric sources[27], uncertainty in the ecosystem model parameters[10], and potentially due to the lack of storms which have been argued to enhance diffusive entrainment of iron from the interior[28]. As a result, iron is the limiting nutrient year round in our simulations, while in the real ACC, silicate limitation is also expected to control diatom growth and transition in phytoplankton community composition[7,8]. Consequently, primary production in our model is biased slightly low, particularly over the summer (Supplementary Note 2, Supplementary Figs. 1 and 2). Dust supply maps indicate a supply of dissolved iron to the Southern Ocean on the order of $O$ (1 μmol Fe m$^{-2}$ yr$^{-1}$) assuming 10% of total aerosol iron is soluble[29]. It is important to note that dust deposition is estimated to account for only about 10% of the overall iron supply in the Southern Ocean, while internal transports make up the rest[30]. Hence, although it would be possible to force our modeled surface iron concentrations to become closer to observations by adding dust, here we focus exclusively on internal transport mechanisms.

The Southern Ocean ecosystem is highly seasonal, with a strong spring bloom occurring between November and January[31,32]. Our model exhibits a strong seasonal cycle, as seen from Fig. 2, which illustrates the simulated climatological seasonal cycle of important physical and biological quantities, averaged over the center of the domain. Our simulations therefore provide a unique opportunity to investigate how seasonality in biological processes interacts with the seasonal cycle in physical transport processes and mixing-layer depth (MLD; definition in Methods section). There is a strong spring bloom, with the vertically integrated phytoplankton biomass ($\langle C_p \rangle$; definition given in Methods section) peaking in early November, after the wintertime MLD has started to shoal (Fig. 2a), consistent with previous characterizations of the spring bloom in the ACC[32]. To characterize the strength of mesoscale and submesoscale turbulence (hereon (sub)mesoscale eddies), we also show the root mean square of vertical velocity ($\overline{w^2}^{1/2}$), which mirrors the MLD

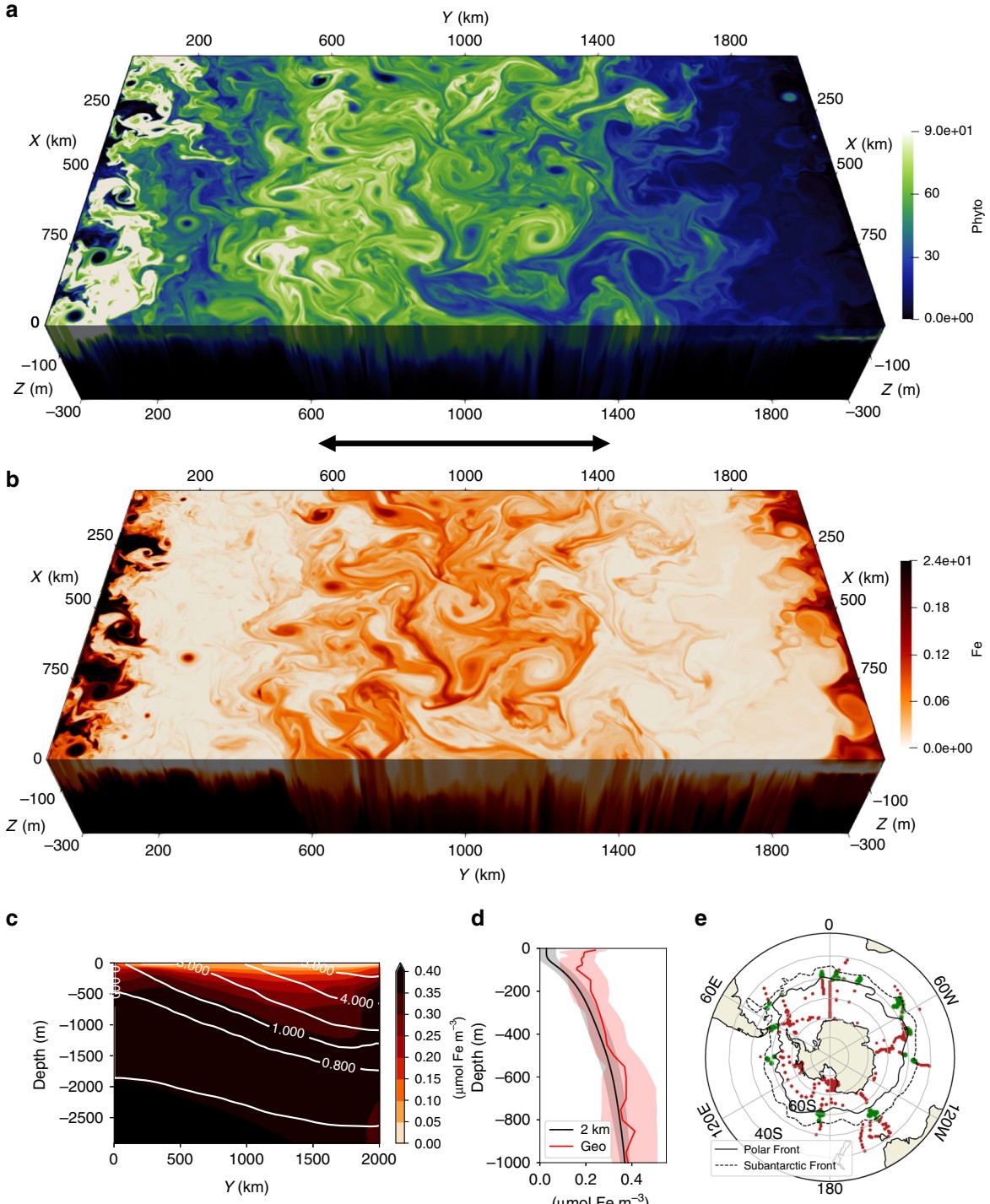

**Fig. 1 A snapshot of phytoplankton biomass in (mg C m$^{-3}$) and iron in (µmol Fe m$^{-3}$) in the top 300 m. a–c** The zonal mean transect of iron and **d** vertical profile averaged over the meridional extent of $y = 600$–1400 km shown as the black arrow in **a** for our 2 km run (black) and median of the GEOTRACES dataset (red) acquired through personal communication with Tagliabue et al.[6] over all profiles in the open ocean region between the climatological position of Polar and Subantarctic front (green; **e**) after applying a three-point median filter in the vertical. The frontal positions were taken from Orsi et al.[50] and extended by 1° to the south and north respectively to incorporate more profiles. The colored shading show the standard deviation for the 2 km run and due to the lack of spatial coverage, the interquartile range is shown for GEOTRACES. The GEOTRACES dataset was biased towards austral summer so the data used in **d** for the 2 km run is over Nov.–Feb.

closely. This suggests that the vertical velocities are associated with mixed-layer instability (MLI), a type of surface-intensified baroclinic instability associated with submesoscales driven by available potential energy within the mixed layer[33], which is more active in winter with deep mixed layers. It is interesting to note that the vertical eddy iron flux ($w'\,Fe'$; where $(\cdot)'$ is defined as the anomaly from the seasonal and zonal climatology using 15-daily snapshot outputs) is in phase with the biomass and not with vertical velocity

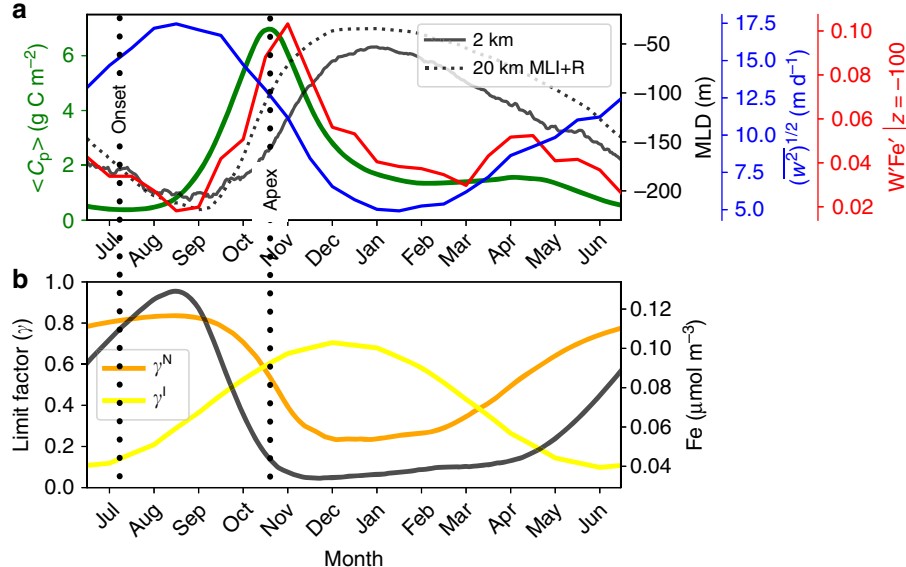

**Fig. 2 Time series of the daily-mean vertically integrated phytoplankton biomass ($\langle C_p \rangle$; green), daily mean of MLD for the 2 km (black solid) and 20 km MLI+R run (black dotted) averaged over the meridional extent of $y$ = 600–1400 km. a**. The seasonal cycle of the root mean square of vertical velocity spatially averaged over the same meridional extent and over 100 m depth from the 2 km run ($\overline{w^2}^{1/2}$) is shown in blue, and spatially averaged vertical eddy iron flux ($w'$Fe' [μmol Fe m$^{-2}$ d$^{-1}$]) at 100 m depth in red. **b** The spatial median over the top 100 m of growth rate limitation factors due to nutrient ($\gamma^N$; orange) and light ($\gamma^I$; yellow) where the former is due to iron year round in our simulation. The iron concentration (Fe) averaged over the top 100 m is plotted in black against the right axis.

itself (Fig. 2a). This suggests that energetic vertical velocities alone are not a sufficient proxy for vertical tracer transport but need to correlate with tracer concentration.

The spring bloom is quantified via $\langle C_p \rangle$, which allows us to define the bloom onset ($\langle C_p \rangle$ minimum) and apex ($\langle C_p \rangle$ maximum) (Fig. 2a[34]). The onset is in late July during the deepening of wintertime mixing layer, and the apex occurs in early November even though surface light conditions ($\gamma^I$; Eq. (S3)) continue to improve over the summer (November–January; Fig. 2b). The decrease in nutrient limitation factor ($\gamma^N$; Eq. (S4)), on the other hand, from 0.8 to 0.2 coincides with the apex and is in phase with iron concentration dropping from 0.13 to 0.03 μmol m$^{-3}$ (Fig. 2b). (The limitation factors ("$\gamma$"s = 0–1) indicate ideal growth conditions when they are unity and zero for no-growth conditions. The effect of grazing by zooplankton is shown in Supplementary Note 3 and Supplementary Fig. 3). This indicates that the decline of the spring bloom in our simulation is due to iron limitation, and not associated with light conditions.

**Vertical eddy and diffusive iron supply for primary production.** To understand what controls the iron concentrations, we now examine the ecosystem in the time–depth plane. The top row of Fig. 3 shows horizontally averaged phytoplankton concentration and vertical iron fluxes by eddies and diffusion vs. time and depth over the seasonal cycle from the 2 km run. Iron concentration is given in Fig. 3f showing signals of wintertime entrainment with the orange contours dipping into the ML around September. (We show the complete zonal-mean iron budget in Supplementary Fig. 1 and time–depth plots of biogeochemical iron consumption in Supplementary Fig. 2.) As in Fig. 2a, there is a strong spring bloom and a mild autumn bloom. Some phytoplankton live below the ML base, particularly during summer when the ML is shallow. During wintertime (July–September) when the ML is deepening and light is low, there is low biomass but high iron concentration (Fig. 3a, f), consistent with the limitation factors ($\gamma^{I,N}$, Fig. 2b).

Iron is supplied to the phytoplankton via three processes: recycling, entrainment, and vertical mixing (here associated with the K-profile parametrization boundary layer; KPP[35]), and vertical eddy fluxes ($w'$Fe'; explicitly resolved by the simulation). Figures 3b, c and S1 show how eddies and vertical (KPP) mixing work together to deliver iron to phytoplankton from depth. Vertical mixing is, by construction, only active within the ML. When vertical gradients of iron are actively sustained by biological consumption (e.g. during the bloom), vertical mixing drives a strong upward diffusive iron flux. This diffusive flux goes to zero at the ML base where KPP turns off. Eddy fluxes, in contrast, peak roughly at the ML base and extend deep into the iron-rich interior, with a magnitude comparable to the diffusive flux in the ML. Thus, eddies play a crucial role in bringing iron across the ML base, where it can be handed off to vertical mixing and delivered to near-surface phytoplankton.

Vertical eddy iron transport is absent from previous estimates of the Southern Ocean iron budget[4–6]. One-dimensional iron budgets suggest that during summer, vertical mixing is not strong enough to supply the iron needed to sustain the observed productivity, implying strong iron recycling within the ecosystem[6]. Our simulations challenge this conclusion, showing that vertical eddy transport can provide a year-round source of iron (Fig. 3b) which exceeds the magnitude of iron recycling (Supplementary Fig. 1).

With the 2 km run as a reference, we use spatial resolution as a parameter to modulate the strength of eddy transport, running two other simulations at eddy-permitting resolutions of 5 and 20 km. The basin-wide density and iron stratification for each resolution are given in Supplementary Fig. 4. Figure 4 shows the annual median of vertically integrated phytoplankton biomass plotted against the annual mean of total (dominated by eddy) vertical iron flux across the ML base, or 100 m, whichever is deeper. This depth scale is chosen to exclude KPP mixing from the flux, and is roughly the depth phytoplankton cease to exist (Fig. 3a, c). As resolution increases from 20 to 2 km for runs

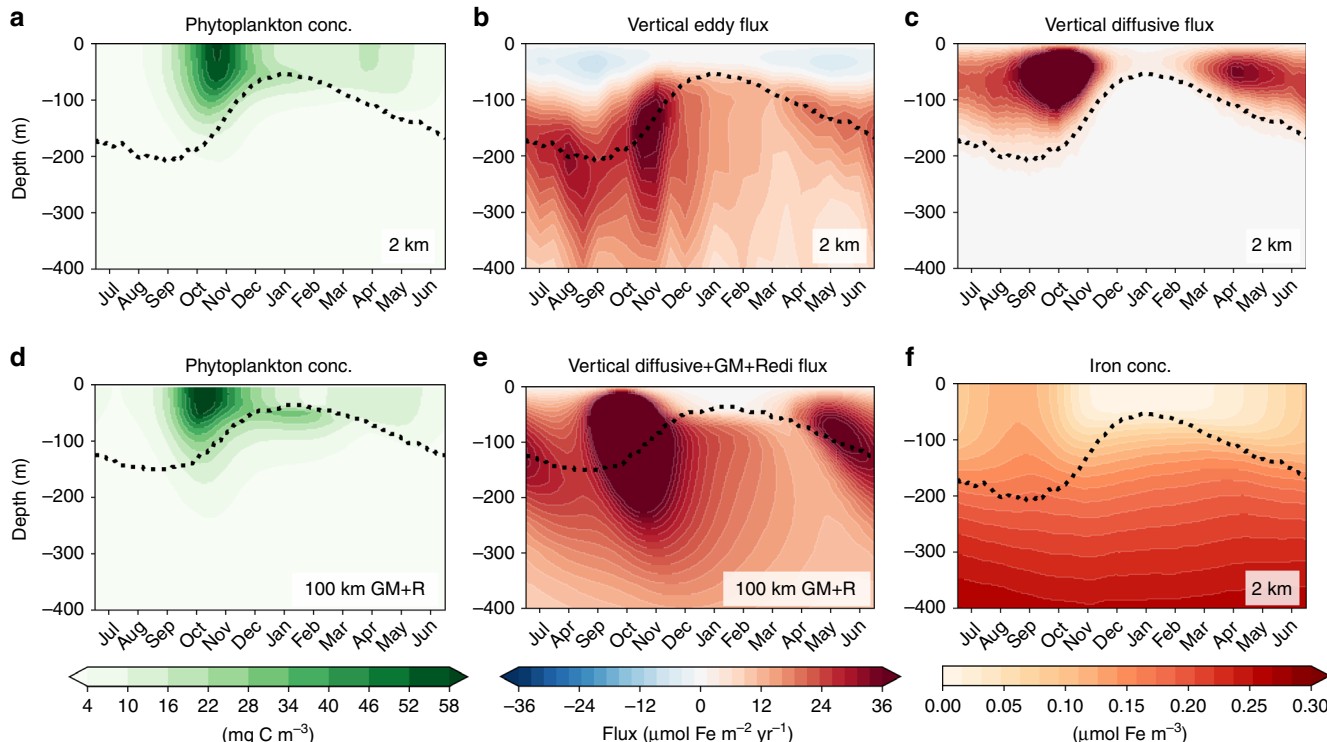

**Fig. 3 Time–depth plots of the daily and spatial median of phytoplankton biomass. a–c** The spatial mean of vertical eddy transport using 15-daily snapshot outputs and diffusive iron transport using daily-averaged outputs. Panels **a–c, f** (daily-averaged iron concentration) are from the 2 km run. **d, e** Daily-averaged phytoplankton biomass and the sum of vertical diffusive, GM and Redi iron flux from the 100 km GM+R run. The dotted lines in all panels show the mixing (mixed) layer depth for the 2 km (100 km GM+R) run. The mixing-layer depth (MLD) was too sensitive to the winds in the 100 km GM+R run, likely due to the GM tapering interacting with KPP[41]. In all of our other runs, the mixed-layer depth defined as the depth at which the potential temperature decreased by 0.2 °C from the surface[49] (not shown) proved to be very similar to the MLD so we used the mixed-layer depth for the 100 km GM+R run.

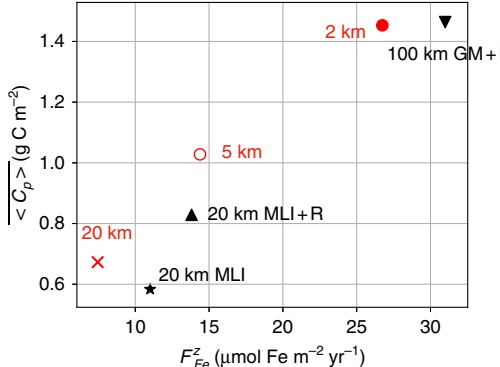

**Fig. 4 A scatter plot showing the resolution and parametrization dependence on annual median phytoplankton biomass ($\overline{\langle C_p \rangle}$) plotted against the annual mean of total vertical iron transport at the ML base or 100 m whichever is deeper ($F^z_{Fe}$).** The runs without any eddy parametrizations are shown in red and the parametrized runs shown in black include the parametrized eddy flux.

without any eddy parametrizations (red markers in Fig. 4), the annual median of daily-averaged phytoplankton biomass ($\overline{\langle C_p \rangle}$) nearly doubles from 0.67 to 1.45 g C m$^{-2}$ in a roughly linear relationship with the annual-mean total (eddy+diffusive) iron transport ($F^z_{Fe}$), which increases from 7 to 27 μmol Fe m$^{-2}$ yr$^{-1}$. This occurs despite a shoaling of the ML, which reduces the entrainment of iron. Thus, in our model ecosystem, eddies effectively control the primary productivity. We show the

time–depth plot of vertical eddy iron flux from each run in Supplementary Fig. 5.

**[Sub]mesoscale eddy parametrizations**. As we move to coarser resolution, we also ask whether conventional eddy parameterizations can provide the missing iron flux. We run three additional simulations at 100 and 20 km resolution. The former represents a standard Coupled Model Intercomparison Project (CMIP)-class ocean GCM, while the latter the newer class of mesoscale-permitting GCMs[36]. Due to limiting computational resources, we will continue to rely on non-eddying and mesoscale-permitting GCMs for global climate and carbon cycle simulations. It is, therefore, informative to examine how commonly employed parametrizations perform compared to submesoscale permitting simulations. The three different eddy parameterizations we experiment with are: Gent-McWilliams' eddy-induced velocity parametrization (GM[37]) in order to represent unresolved mesoscale restratification in the interior, isopycnal tracer diffusion (Redi[38]) to represent mesoscale stirring of tracers, and MLI parametrization[39] to represent the shoaling of ML due to otherwise resolved MLI. The runs are: 100 km GM+R, 20 km MLI+R, and 20 km MLI. The first case is with GM and Redi at 100 km resolution. We allowed the GM coefficient to vary between 200 and 2500 m$^2$ s$^{-1}$, depending on the vertical-mean Richardson number[40], and chose a tapering scheme which accounted for a smooth transition between the diabatic boundary layer and adiabatic interior[41]. The Redi diffusivity was chosen as 1000 m$^2$ s$^{-1}$. The 20 km MLI+R run is with MLI and Redi at 20 km resolution. We tuned the MLI parameters to produce the same wintertime MLD as the 2 km simulation (Fig. 2a, black

dashed curve). The Redi diffusivity was chosen as $200 \, \mathrm{m^2 \, s^{-1}}$, smaller than the case above with mesoscale eddies partially resolved at 20 km resolution. The 20 km MLI run is with only the MLI parametrization at 20 km resolution. Further details on each configuration are given in Supplementary Note 4.

The parameters in eddy parametrizations in global climate simulations are chosen operationally, without community-wide established best practices. In our study, we performed extensive experimentation with different combinations of eddy parameterizations and parameters, here reporting only the most relevant results. We discovered that, with a novel combination of choices, the parametrized eddy flux in 100 km GM+R run captures the amplitude and timing of cross-ML-base vertical eddy iron transport, particularly around November as seen in the 2 km run (Fig. 3b, e). The vertical flux in the 100 km GM+R is the sum of KPP mixing, GM advection, and Redi diffusion. In other words, a direct comparison of vertical iron flux between the 100 km GM+R and 2 km run is Fig. 3e against the sum of panels b and c in Fig. 3. Although not shown, the resolved eddy advection contribution is negligible at 100 km resolution and vertical mixing (KPP) is contained within the ML. The cross-ML-base iron transport in Fig. 3e is, therefore, predominantly due to isopycnal eddy stirring. Setting the Redi diffusivity to zero—equivalent to no mesoscale isopycnal stirring—in the 100-km run results in $F^z_{\mathrm{Fe}}$ decreasing by a factor of two and vertically integrated annual phytoplankton biomass by ~40%. The pulse of iron coincides roughly with the spring bloom apex in early November in both the 2 and 100 km GM+R runs, but summertime (January–March) biomass is higher within the top 100 m in the former (Fig. 3a, d). The higher summertime biomass in the 2 km run may be due to partially resolved MLI actively generating vertical iron gradients within the top 100 m, allowing for larger diffusive flux in the top 100 m for the 2 km run than in the 100 km GM+R run (Fig. 3c, e).

We plot phytoplankton biomass against vertical iron transport also for the parametrized runs in Fig. 4. Consistent with Fig. 3, they remain similar between the 2 and 100 km GM+R runs (Fig. 4). The 20 km MLI+R comes close to the 5 km run (Fig. 4) with Redi diffusion adding cross-ML-base iron transport (Supplementary Fig. 6). The MLI parametrization contribution, intended to replicate the restratification of the ML and not eddy tracer transport[39], is contained within the ML and does not enhance cross-ML-base iron transport (Supplementary Fig. 6b). Isopycnals, and consequently iron contours, in the interior at 20 km resolution are too steep compared to the 2 km run, with insufficient restratification relative to the resolved-mesoscale run (Supplementary Figs. 4c and 7b). This results in weaker vertical gradients of iron and less net iron supply via entrainment and vertical eddy transport. The GM parametrization in the 100 km resolution run allows us to improve isopycnal steepness (Supplementary Figs. 4d and 7c), and the Redi diffusivity is used to tune the isopycnal iron transport. The 20 km MLI run performs the worst among the parametrized runs (black markers in Fig. 4) with cross-ML-base eddy iron transport coming only from the resolved eddies at 20 km resolution.

## Discussion

We have shown, using a configuration representing the zonal-mean view of the Antarctic Circumpolar Current region, that eddy iron transport is crucial in supplying iron from depths across the mixing-layer base (Figs. 3 and 4). A study using a similar zonally re-entrant channel model, also found an increase in wintertime (August–October) vertical eddy iron transport, and consequently elevated primary production during September–October[28]. Their spatial resolution of 1/24°, however, is similar to our 5 km run and the relative contribution of eddy transport in their study is likely

underestimated (Fig. 4). Although 2 km resolution is state-of-the-art for a basin-scale simulation coupled to a full biogeochemical model, it is not sufficient to fully resolve submesoscale processes including MLI[15]. Based on the resolution dependence, we would expect the role of eddies in supplying iron to increase further with higher resolutions[42], but this would only strengthen the central finding that eddy iron transport modulates primary productivity in the open Southern Ocean.

Our results suggest that, in order to adequately capture the eddy iron transport, we should either at least partially resolve the submesoscales (2 km run) or completely parametrize the eddies using the current generation of GM (100 km GM+R run). In particular, we found that a novel combination of the Visbeck scheme for scaling the GM coefficient based on linear baroclinic instability[40], combined with the Ferrari tapering method[41], was uniquely able to mimic the eddy fluxes from the high-resolution run. Looking forward, it would be interesting to see whether recently developed energy backscattering GM parametrizations[43,44] would improve tracer transport in mesoscale-permitting models. The agreement of the 100 km GM+R run with the 2 km run, however, also highlights the potential significance of improving the parametrization for mesoscale isopycnal tracer (Redi) diffusion, which has been argued to be a significant factor in tracer ventilation using ship-track observations in the Southern and Arctic Oceans[45]. In our study, the Redi diffusivity was tuned in an ad hoc manner; future eddy parameterizations instead must be able to determine the correct value of this parameter based on physics in order to accurately simulate the response of the Southern Ocean biological pump to climate change. Considering that the MLI parametrization in its current formulation, intended for density restratification, does not capture eddy tracer transport (Supplementary Fig. 6), it may also be beneficial to develop a new parametrization for the effects of submesoscale isopycnal tracer stirring.

There has been growing evidence regarding the relative importance of eddies in the biological carbon pump[46–48]. The eddies responsible for supplying iron also export phytoplankton downwards in our simulation. We show in Supplementary Fig. 8a the time–depth plot of vertical eddy phytoplankton transport ($w'C'_\mathrm{p}$) for the 2 km run. The eddies subduct phytoplankton across the ML base and the magnitude increases with resolution (Supplementary Fig. 8b). Nevertheless, the annual phytoplankton biomass and primary production increase with resolution (Figs. 4 and Supplementary Fig. 8b), indicating that the eddy supply of iron and resulting increase in productivity overcompensate for the loss of phytoplankton due to eddy subduction. Considering the annual maximum of eddy subduction occurs after the annual maximum in primary production, accurate representation of the magnitude and timing of eddy carbon subduction may be necessary to accurately model the Southern Ocean carbon cycle.

## Methods

**Mixing layer.** The MLD is the boundary layer over which isotropic turbulent mixing, parameterized by the KPP in this simulation, is enhanced. Here, we quantify the depth of this highly variable layer as the zonal 99th percentile of the daily-averaged KPP boundary layer. In our simulations, the *mixed*- and *mixing*-layer depth tended to be similar to each other. In general, however, the *mixed* layer often used in observational studies can be deeper than the *mixing* layer as the former is defined purely by thermal dynamical properties[49] while as latter is defined by kinematic properties. We argue that the *mixing* layer is the relevant depth scale for tracer transport as it is the layer over which diapycnal mixing is active[15]. Figure 3c shows that diffusive fluxes are only active within the mixing layer in our simulation when eddies are explicitly resolved.

**Integrated phytoplankton biomass.** The integrated biomass ($\langle C_\mathrm{p} \rangle$) is defined as the full-depth vertical integration of the spatial median ($y = 600–1400$ km, $x = 0–1000$ km) of $C_\mathrm{p}$ in order to incorporate phytoplankon existing below the mixing layer[5]. We take the median as the phytoplankton biomass in our model approximately has a log-normal distribution.

**Reporting summary**. Further information on research design is available in the Nature Research Reporting Summary linked to this article.

## Data availability
The simulation outputs for 15-daily snapshot and monthly-averaged outputs of physical variables ($v$, $\theta$, $\Phi$) are available on Pangeo (https://github.com/pangeo-data/pangeo-datastore/blob/master/intake-catalogs/ocean/channel.yaml). Correspondence and requests for other variables and materials should be addressed to the leading author (email: takaya@ldeo.columbia.edu)

## Code availability
The model configuration is available on Github (https://doi.org/10.5281/zenodo.3266400). Example Jupyter notebooks used for our spectral analysis are available on Pangeo (https://doi.org/10.5281/zenodo.3358021).

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

## Acknowledgements

This research was supported by NASA Award NNX16AJ35G as part of the SWOT Science Team. R.A. acknowledges additional support from NSF Awards OCE-1553593 and OCE-1740648. M.L. acknowledges additional support from CNES and ANR award (ANR-16-CE01-0014).

## Author contributions

T.U. ran and analyzed all of the simulations provided in this study. D.B. helped setting up the physical configuration. R.A. provided overall guidance as the Ph.D. mentor of T.U. and configuring the eddy parametrizations. G.M. provided guidance in setting the parameters in the biogeochemical model. M.L. and S.S. provided insight into the biogeochemical and physical outputs, respectively.

## Competing interests

The authors declare no competing interests.
