## [Peer Review File · Nature Communications]

**REVIEW OF: 'EDDY IRON FLUXES CONTROL PRIMARY PRODUCTION
IN THE OPEN SOUTHERN OCEAN'**

This manuscript examines the role of eddy vertical fluxes, and vertical turbulent fluxes, in the supply of iron to the surface ocean mixed layer in numerical simulations of a re-entrant channel, intended to represent an idealized version of the Southern Ocean. This problem is motivated by the need to understand how iron, a limiting nutrient for ocean primary productivity in the Southern Ocean, is supplied to the surface mixed-layer. As reviewed in the manuscript, much existing work has tended to emphasize the role of surface fluxes (ie. through deposition), or 1D dynamics (ie. entrainment/diffusion), however here the role of eddies in bringing iron rich water up from below the surface mixed-layer is emphasized. The topic is important, with implications for understanding, and modeling, the ocean's role in carbon uptake.

The analysis of the manuscript appears to be well designed and a convincing story is laid out as to the role of mesoscale eddies in bringing nutrients from the base of the mixed-layer up to where turbulent mixing (parameterized) can further mix it up into the euphotic zone. However, I think there are a number of issues that make it not the best fit for Nature. These are detailed below, but to summarize, first the maximum resolution of the model (2km) likely does not resolve enough of the submesoscale in this region to support the conclusions related to this topic. Next, a significant portion of the manuscript relies on the use of several eddy parameterizations, which, while interesting and important to the ocean modeling community, are not fully explained and may not be of sufficiently broad interest for Nature. Finally, the extent to which the findings from the idealized channel configuration will translate to the real Southern Ocean is unknown, leaving some of the manuscript's claims unsupported.

1. MAJOR COMMENTS

- (1) The title of the manuscript implies that the results prove that the mechanism identified here (eddy vertical nutrient fluxes at the base of the mixed-layer) is the dominant contributing term in the Southern Ocean iron budget. This however does not seem to be fully supported by the analysis, which excludes other mechanisms such as surface dust deposition. Likewise, the comparison with data is fairly thin (figure 1d), and actually seems to indicate significant differences between the modeled Fe distribution and observations (particularly near the surface). Given these limitations, this manuscript reads more like a process study paper, focused on the interaction between vertical eddy fluxes and mixed-layer turbulence, with implications for recent 1D theories of nutrient entrainment, and practical implications for ocean modeling, but without a very direct line to the overall importance in Southern Ocean primary productivity.
- (2) One of the conclusions of the manuscript is that the role of submesoscale surface mixed-layer instability (MLI) is of limited importance (eg. line 14-15 in the abstract). However, as eventually noted in the conclusions, the finest resolution of 2km does not resolve MLI (line 208). This could also be noted using the wavelength of the fastest growing linear mode (see for example Fox-Kemper et al., 2008):

$$L = \frac{2\pi U}{|f|} \sqrt{\frac{1 + Ri}{5/2}},$$

which would give a wavelength close to the Nyquist wavelength for $U \sim 0.1$ m/s and $Ri = 1$. The arguments then in the manuscript about the role of the submesoscale all hinge on the MLI parameterization runs, which is not necessarily an entirely fair

comparison, particularly as the parameterization overturning streamfunction may not entrain as much as the resolved instabilities would (Fox-Kemper and Ferrari, 2008). It is also noted that the MLI parameterization is not intended to capture tracer fluxes (supplementary information), but that is never discussed in relation to the conclusions of the manuscript.

It would also be important to put these results in the context of recent work on the role of mesoscale and submesoscale instabilities in high-resolution realistic models of the Southern Ocean. For instance, Bachman et al. (2017) found that there was a strong regional dependence on the prevalence of submesoscale MLI, and likewise that going from a nominal 2km resolution to nominal 500 m resolution ($1/198^\circ$) led to a strong increase in the amount of vertical velocity variance, due to resolved MLI. This makes it difficult to evaluate how robust the findings here are as to the role, or lack thereof, of the submesoscale in bringing nutrients up from depth in the Southern Ocean (connecting here again to comment 1 above).

- (3) A major portion of the results, and interpretation of the physical processes at play, hinge on the comparison between the various model resolutions and different parameterizations. This portion of the work was well done, but read a bit technical for Nature. For instance, interpreting figure 4 hinges on the reader understanding what physics are parameterized by Reddi diffusion, GM advection, and the MLI parameterization. It would not be possible to explain these satisfactorily within the restrictions of a Nature manuscript, but I am also not confident that the results can be adequately understood without this background.

2. MINOR COMMENTS

- (1) The ‘(sub)mesoscale’ notation is not standard, and never explained. In general I would suggest not using this notation, however if you do please don’t use it in the abstract without explanation.
- (2) It is preferable to use a dynamical definition of submesoscale (ie. $O(1)$ Rossby and Richardson numbers), as opposed to a qualitative length scale (eg. line 39). Particularly in this region where the deformation radius is $O(10)$ km). See for instance McWilliams (2016).
- (3) Line 90: ‘...our simulations are able to reproduce this seasonal cycle...’ This is trivially true, as the seasonal cycle being referred to is one derived from the model (not for instance from an observational climatology).
- (4) Figure 2b: Put ‘Fe’ in the legend.
- (5) There appears to be an extra, blank, reference (15) in the supplementary information. The references past this don’t match with the numbers given in the text.

REFERENCES

- Bachman, S. D., J. R. Taylor, K. A. Adams, and P. J. Hosegood, 2017: Mesoscale and Submesoscale Effects on Mixed Layer Depth in the Southern Ocean. *Journal of Physical Oceanography*, **47** (9), 2173–2188, doi:10.1175/JPO-D-17-0034.1, URL <http://journals.ametsoc.org/doi/10.1175/JPO-D-17-0034.1>.
- Fox-Kemper, B., and R. Ferrari, 2008: Parameterization of Mixed Layer Eddies. Part II: Prognosis and Impact. *Journal of Physical Oceanography*, **38** (6), 1166–1179, doi:10.1175/2007JPO3788.1, URL <http://journals.ametsoc.org/doi/abs/10.1175/2007JPO3788.1>.

- Fox-Kemper, B., R. Ferrari, and R. Hallberg, 2008: Parameterization of Mixed Layer Eddies. Part I: Theory and Diagnosis. *Journal of Physical Oceanography*, **38** (6), 1145–1165, doi:10.1175/2007JPO3792.1, URL <http://journals.ametsoc.org/doi/abs/10.1175/2007JPO3792.1>.
- McWilliams, J. C., 2016: Submesoscale currents in the ocean. *Proceedings of the Royal Society A: Mathematical, Physical and Engineering Science*, **472** (2189), 20160117, doi:10.1098/rspa.2016.0117, URL <http://rspa.royalsocietypublishing.org/lookup/doi/10.1098/rspa.2016.0117>.

Reviewer #2 (Remarks to the Author):

Review

The paper used idealised simulations of a channel to represent the Southern Ocean and explore the role submesoscale processes play in the supply of iron to the euphotic zone. The study showed submesoscale processes vertically transport iron into the water just below the mixed layer where it is more efficiently fluxed into the euphotic zone and fuels increased phytoplankton. The study is important and supports previous Lagrangian simulations and observational work on the importance of vertical eddy transport of iron supply to the euphotic zone.

While I'm supportive of publishing the paper, I have a few issues to address.

1. the paper chooses to show the response of phytoplankton to variation in iron supply. However, the phytoplankton response is sensitive to zooplankton grazing - i.e. could vary from no response if the grazing controls growth to a big response. How does primary productivity and export production change in the considered simulations and do these scale like the depth-integrated phytoplankton?

2. The study focused on how resolving the mesoscale eddies increased vertical eddy supply of iron to the euphotic zone, but it may also cause greater export of phytoplankton out of the euphotic zone. How do resolving eddies alter the export of phytoplankton and organic matter out of the euphotic zone?

3. Figure 3 shows the RMS of the vertical velocity, which is related to the MLD instability, while this is interesting, the text suggests this is not an important term. You should show the vertical eddy transport at say 100 m to provide a visual measure of how the transport changes over the season. Further, I would like to see the vertical eddy transport from all simulations. How do the seasonal transports for cases when eddies are resolved compared to the cases when eddies are not resolved? What does it imply for how one parameterises eddies in coarse resolution simulations.

specifics

I97 why show the vertical velocity RMS if MLI is not the dominant term in the vertical eddy flux term? Show the vertical eddy flux at 100m.

I 125 Not clear that vertical mixing is referring to Kpp mixing

I132 How are the vertical eddy fluxes computed? It would be useful to show the fluxes from all experiments

I151 is 2km resolution sufficient? How much more resolution?

I153 - suggest lower -> coarser

I179 - and mean integrated phytoplankton by how much?

I179 - Delete "In other words" and start a new paragraph

I213- interesting that the coarse resolution with GM+R can be as good as the 2km model at supplying iron to the euphotic zone - no need for more resolution. What other tracers could be used to help tune the coarse resolution model?

Introduction

We thank both reviewers for their positive and constructive feedback. We have acknowledged their contribution in the Acknowledgements section (line 363). Please find our revisions to the manuscript shown in red text in the file: "Changes-to-manuscript.pdf."

Reviewer #1

Major comments:

The title of the manuscript implies that the results prove that the mechanism identified here (eddy vertical nutrient fluxes at the base of the mixed-layer) is the dominant contributing term in the Southern Ocean iron budget. This however does not seem to be fully supported by the analysis, which excludes other mechanisms such as surface dust deposition. Likewise, the comparison with data is fairly thin (Figure 1d), and actually seems to indicate significant differences between the modeled Fe distribution and observations (particularly near the surface). Given these limitations, this manuscript reads more like a process study paper, focused on the interaction between vertical eddy fluxes and mixed-layer turbulence, with implications for recent 1D theories of nutrient entrainment, and practical implications for ocean modeling, but without a very direct line to the overall importance in Southern Ocean primary productivity.

We appreciate the reviewer's perspective. We essentially agree that our paper is a process study; however, process studies have long been a crucial part of scientific progress in oceanography, and we don't feel this should preclude publication in any particular journal. The key question is whether the results constitute a significant advance. Our study is the first ever to run such a high resolution physical model with a fully dynamic ecosystem in a Southern Ocean context, and we feel there is much of relevance to be learned here. We feel that our general conclusion regarding the central importance of eddies in the iron budget will stand the test of time.

Regarding the comparison with data, it is important to note how much closer our iron profile is to the observations than global-scale climate models (Tagliabue et al., 2016). Achieving such good agreement required lots of work! Nevertheless, as the reviewer pointed out, the near-surface iron concentration in our model is lower than observed from ship-track measurements. We attribute this to us not including external sources such as dust in our simulation. This idealization was deliberate in order not to conflate the effects of iron supply by eddies with other sources. The discrepancy of iron concentration is within the range of values estimated from dust maps. To clarify this limitation, we have added in the manuscript:

“Dust supply maps indicate a supply of dissolved iron to the Southern Ocean on the order of $O(1 \mu \text{mol Fe m}^{-2} \text{ yr}^{-1})$ assuming 10% of total aerosol iron is soluble (Duce et al., 1991). Hence, although it would be possible to force our modelled surface iron concentrations to become closer to what is observed by adding dust, here we focus exclusively on internal transport mechanisms.” (lines 87-90).

It is important to note that dust deposition accounts for only about 10% of the overall iron supply in the Southern Ocean, while internal transports make up the rest (Sarmiento 2013). On this basis, we expect that the conclusions from our process model are indeed applicable to the overall Southern Ocean iron budget. However, to address the reviewer’s point, we have changed the title to “Vertical Eddy Iron Fluxes Support Primary Production in the Open Southern Ocean” and softened some of our claims in the conclusion (lines 224-226).

One of the conclusions of the manuscript is that the role of submesoscale surface mixed-layer instability (MLI) is of limited importance (e.g. line 14-15 in the abstract). However, as eventually noted in the conclusions, the finest resolution of 2km does not resolve MLI (line 208). The arguments then in the manuscript about the role of the submesoscale all hinge on the MLI parameterization runs, which is not necessarily an entirely fair comparison, particularly as the parameterization overturning streamfunction may not entrain as much as the resolved instabilities would (Fox-Kemper and Ferrari, 2008). It is also noted that the MLI parameterization is not intended to capture tracer fluxes (supplementary information), but that is never discussed in relation to the conclusions of the manuscript.

As the reviewer surely knows, in ocean modeling there is never enough resolution to completely resolve all processes of interest. Our runs are groundbreaking because they are the highest resolution simulations to date with a fully dynamic ecosystem in a Southern Ocean context. While MLI is not fully resolved at 2km, it is certainly permitted. We showed this in an earlier manuscript (Uchida et al., 2017) via linear stability analysis and spectral energy budgets; MLI can be partially resolved even at 10km resolution! For these simulations, further evidence regarding the role of the submesoscale vs mesoscale comes from spectral analysis of the simulations, as reported in a separate paper (Uchida et al., 2019; JAMES).

Nevertheless, we agree that our results may be underestimating the relative role of tracer transport associated with mixed-layer instability (MLI) due to MLI only being partially resolved at 2km resolution. We have, therefore, removed the claim “..., rather than mixed-layer instability” from the abstract. We have also added in the conclusions regarding the MLI parametrization:

“Considering that the MLI parametrization in its current formulation, intended for density restratification, does not capture eddy tracer transport (Fig. S5), it may be beneficial to add the effects of submesoscale isopycnal tracer stirring” (lines 237-240).

It would also be important to put these results in the context of recent work on the role of mesoscale and submesoscale instabilities in high-resolution realistic models of the Southern Ocean. For instance, Bachman et al. (2017) found that there was a strong regional dependence

on the prevalence of submesoscale MLI, and likewise that going from a nominal 2km resolution to nominal 500 m resolution (1/198) led to a strong increase in the amount of vertical velocity variance, due to resolved MLI. This makes it difficult to evaluate how robust the findings here are as to the role, or lack thereof, of the submesoscale in bringing nutrients up from depth in the Southern Ocean (connecting here again to comment 1 above).

Figure 4 indicates that we have not reached numerical convergence at 2km resolution, even though this resolution is the current state of the art for a basin-scale ocean simulation coupled to a full biogeochemical model. As the reviewer points out, we would expect a further increase in eddy iron transport with spatial resolution. This would, however, only strengthen our central claim that eddies are of first-order importance in the open Southern Ocean iron budget. We have added a discussion of this point, and a reference to Bachman et al. (2017) in lines 224-226.

A major portion of the results, and interpretation of the physical processes at play, hinge on the comparison between the various model resolutions and different parameterizations. This portion of the work was well done, but read a bit technical for Nature. For instance, interpreting Figure 4 hinges on the reader understanding what physics are parameterized by Redi diffusion, GM advection, and the MLI parameterization. It would not be possible to explain these satisfactorily within the restrictions of a Nature manuscript, but I am also not confident that the results can be adequately understood without this background.

Citing Nature Communications' mission statement: "Nature Communications is an open access journal that publishes high-quality research from all areas of the natural sciences. Papers published by the journal represent important advances of significance to specialists within each field." - <https://www.nature.com/ncomms/about>. We, therefore, believe that our results on eddy parametrizations are significant to people in the biogeochemical community interested in modelling primary production in the open Southern Ocean, as well as to ocean modellers and the entire climate modelling community, since primary production is linked to the biological carbon pump.

We have added the rationale for examining the performance of eddy parametrizations in lines 166-169.

Minor comments:

The '(sub)mesoscale' notation is not standard, and never explained. In general I would suggest not using this notation, however if you do please don't use it in the abstract without explanation.

We have added the definition in the abstract as: "mesoscale and submesoscale turbulence (hereafter (sub)mesoscale eddies)" (line 11).

It is preferable to use a dynamical definition of submesoscale (ie. $O(1)$ Rossby and Richardson numbers), as opposed to a qualitative length scale (e.g. line 39). Particularly in this region where the deformation radius is $O(10 \text{ km})$. See for instance McWilliams (2016).

We have added: "... and associated with Rossby numbers on the order of unity" (line 40).

Line 90: '...our simulations are able to reproduce this seasonal cycle...' This is trivially true, as the seasonal cycle being referred to is one derived from the model (not for instance from an observational climatology).

What we intended to point out was that our modelled seasonal cycle is in agreement with satellite observations. We have changed the expression to:

"The Southern Ocean ecosystem is highly seasonal, with a strong spring bloom occurring between November and January (Sallee et al., 2015, Ardyna et al., 2017). Our model exhibits a strong seasonal cycle, as seen from Fig. 2, which illustrates the simulated climatological seasonal cycle of important physical and biological quantities, averaged over the center of the domain. Our simulations therefore, provide..." (lines 93-96).

Figure 2b: Put 'Fe' in the legend.

As iron concentration is plotted against the y axis on the right of the figure, separate from the growth limitation factors, we have added "Fe" in the caption of the figure.

There appears to be an extra, blank, reference (15) in the supplementary information. The references past this don't match with the numbers given in the text.

Reference (15) was a typo and we have removed it.

Reviewer #2

Major comments:

The paper chooses to show the response of phytoplankton to variation in iron supply. However, the phytoplankton response is sensitive to zooplankton grazing - i.e. could vary from no response if the grazing controls growth to a big response. How does primary productivity and export production change in the considered simulations and do these scale like the depth-integrated phytoplankton?

We agree with this point that phytoplankton phenology is sensitive to zooplankton grazing. We show in Fig. S3c (supplementary information) the zooplankton biomass in our model. The grazing rate is a parameter poorly constrained by observations, but our results suggest that iron

limitation is the dominant factor, rather than grazing, in regulating our modelled phytoplankton seasonality.

The study focused on how resolving the mesoscale eddies increased vertical eddy supply of iron to the euphotic zone, but it may also cause greater export of phytoplankton out of the euphotic zone. How do resolving eddies alter the export of phytoplankton and organic matter out of the euphotic zone?

This is an excellent point, and we thank the reviewer for raising it. We have investigated this and added Fig. S8a (reproduced below). The figure shows the vertical eddy transport of phytoplankton ($w'C'_p$) from the 2km run. As the reviewer speculates, the eddies responsible for upward iron transport indeed transport phytoplankton downwards across the mixing layer base. Nevertheless, the overall biomass increases with resolution as shown from Fig. 4 in the main text, indicating that the iron supplied by eddies, and the resulting increase in productivity, overcompensate for the loss by downward eddy export of phytoplankton.

The resolution dependence of vertically integrated primary production ($\langle PP \rangle$) and eddy transport ($w'C'_p$) is also shown below (Fig. S8b). The eddy transport is shown at the depth of ML base or 100m whichever is deeper. The spring production peaks earlier and the baseline productivity increases for higher resolution runs due to more iron available in the surface ocean by the eddy supply and the eddy export of phytoplankton occurs slightly after the annual maximum of production (lines 241-250).

Figure 3 shows the RMS of the vertical velocity, which is related to the ML instability, while this is interesting, the text suggests this is not an important term. You should show the vertical eddy transport at say 100 m to provide a visual measure of how the transport changes over the season. Further, I would like to see the vertical eddy transport from all simulations. How do the seasonal transports for cases when eddies are resolved compared to the cases when eddies are not resolved? What does it imply for how one parameterizes eddies in coarse resolution simulations.

We have addressed this in supplementary Fig. S5, also attached below. The figure shows time-depth plots of vertical eddy iron flux from the runs with no eddy parametrizations. The MLD

shoals and the overall amplitude of vertical eddy iron transport increases with spatial resolution. In particular, the 20 km run misses the strong pulse of iron transport around November. Our results suggest that for runs with eddy permitting resolutions (e.g. 20 km in our case), we should at least incorporate the effect of along-isopycnal tracer transport.

Minor comments:

Line 97: Why show the vertical velocity RMS if MLI is not the dominant term in the vertical eddy flux term? Show the vertical eddy flux at 100m.

We show the RMS of vertical velocity as one indicator of our model having seasonality in the physics, but have added the vertical eddy iron flux to Fig. 2, shown in red (lines 106-109).

Line 125: Not clear that vertical mixing is referring to KPP mixing.

We have changed it to: "vertical (KPP) mixing" (line 134).

Line 132: How are the vertical eddy fluxes computed? It would be useful to show the fluxes from all experiments?

The vertical eddy fluxes are calculated as the difference between the snapshot outputs and zonal seasonal mean of each variable. We have added: "... vertical eddy fluxes ($w'Fe'$; explicitly resolved by the simulation)." (lines 133). The definition is added in lines 106-109. We also show the eddy fluxes from the non-parametrized runs in Fig. S5.

Line 151: Is 2km resolution sufficient? How much more resolution?

It is likely that 2km is not sufficient to fully resolve MLI or the inverse energy cascade associated with it. However, as discussed in reply to Reviewer #1, 2km resolution should capture a substantial fraction of it. We speculate that once the spatial resolution becomes sufficiently fine to resolve the forward cascade of kinetic energy, we would reach numerical convergence in the physics and tracer transport.

Line 153: suggest "lower" -> "coarser"

We have adopted your suggestion (line 163).

Line 179: And mean integrated phytoplankton by how much?

The vertically integrated annual phytoplankton biomass decreases by roughly 40% (lines 194-195).

Line 179: Delete "In other words" and start a new paragraph

We have moved the sentence to lines 189-190.

Line 213: Interesting that the coarse resolution with GM+R can be as good as the 2km model at supplying iron to the euphotic zone - no need for more resolution. What other tracers could be used to help tune the coarse resolution model?

Although the 100 km GM+R performs well, it required extensive tuning due to the parameters in the parametrizations being poorly constrained by physical insight. The caveat here is that our tuned parameter values may not directly carry over to other basins, and so great caution should be used in extrapolating its effects on simulations of a future warming climate. As we mention in our Discussion, future work should require the GM, Redi and MLI parameters to be physically informed, rather than be dependent on the model configuration and tuned in an ad-hoc manner.

One way to better constrain the Redi parameter could be to have multiple synthetic passive tracers initialized orthogonally to each other in the simulation and generate a best fit estimate (e.g. Abernathey et al., 2013).

Reviewers' comments:

Reviewer #1 (Remarks to the Author):

REVIEW OF: 'VERTICAL EDDY IRON FLUXES SUPPORT PRIMARY PRODUCTION IN THE OPEN SOUTHERN OCEAN'

In the prior review my two principal concerns were:

- (1) The results pertaining to the role of the submesoscale were not well supported.
- (2) The manuscript may not be of sufficient interest for Nature.

Regarding the first point, the authors have satisfactorily addressed my concerns by rewording the manuscript to not emphasize comparisons between the fully-resolved mesoscale and the partially-resolved submesoscale.

On the second point I still have reservations. My original critique was that the comparison between the parameterized eddy runs might be too specialized to warrant publication in Nature. Since that last review the authors have also published a new paper that appears to have some level of overlap with this manuscript (Uchida et al., 2019). In particular, it appears that the 'central finding that eddies provide an efficient pathway for iron supply in the open Southern Ocean' (Line 225 this manuscript) is also a major focus of the other paper, which uses a similar (same?) model to argue that eddy transport is of 'first-order importance for iron supply from the interior to the surface' (Uchida et al., 2019, key point 2 of that manuscript). The above is not to suggest that the authors are unfairly duplicating findings, rather it is meant to suggest that the section on eddy parameterizations might be the most important new result of this particular manuscript. The authors argue in their response that the parameterization section is of sufficient interest to the biogeochemical and climate modeling communities to justify publication. These topics are outside my area of expertise, so it is difficult for me to evaluate whether this manuscript meets the criteria of providing a significant enough advance for publication in Nature.

Aside from the above concern, the analysis appears to be well constructed and executed, the writing is clear, and I feel the manuscript is suitable for publication.

1. MINOR COMMENTS

- (1) Line 40: Generally the submesoscale is dynamically defined as having both Rossby *and* Richardson numbers of order unity.
- (2) Line 163: sp. *coarser*
- (3) Around Line 172: You currently give short explanations of the physics captured by each parameterization in the bulleted list, but I think it would help the reader if you brought the short physical descriptions out to where you first introduce the 3 runs you perform.

REFERENCES

Uchida, T., D. Balwada, R. Abernathey, G. McKinley, S. Smith, and M. Lévy, 2019: The contribution of submesoscale over mesoscale eddy iron transport in the open Southern Ocean. *Journal of Advances in Modeling Earth Systems*, 2019MS001805, doi:10.1029/2019MS001805, URL <https://onlinelibrary.wiley.com/doi/abs/10.1029/2019MS001805>.

Reviewer #2 (Remarks to the Author):

Review

The revised paper has addressed my comments and I recommend publication.

I just have a few minor comments that authors should consider.

l107-109 - An interesting statement but what does it imply? MLI is not important? Make some statement otherwise it is not clear why this sentence is needed.

l188 -192 - I suggest you remove the KPP term from the simulated iron flux in the 100km RM+R run to make it easier to compare to the 2km simulation that permits sub-mesoscale stirring. I think this change along with your statement the KPP mixing in the 100km GM+R is confined to the ML (consistent with the 2km run) would be sufficient.

l225-226 - Any thoughts on how much more Fe supply could increase if one resolved sub mesoscale processes?

l250 - I note that eddy export of Phyto is less than 15% of the simulated PP. This term increases substantially with resolution. Importantly, there is a spatial separation of the eddy iron supply into the ML from the eddy phyto export from the ML. This spatial and temporal separation enables the ecosystem to cycle the iron in a manner that reduces the fraction eddy Fe supply to eddy Fe export.

Introduction

We thank both reviewers for their thorough reading and approval of our revisions. Please find our revisions to the manuscript in the file: "Changes-to-manuscript.pdf" and the line numbers below correspond to the lines in this document.

Reviewer #1

Major points

... it appears that the '*central finding that eddies provide an efficient pathway for iron supply in the open Southern Ocean*' is also a major focus of the other paper, which uses the same model to argue that eddy transport is of '*first-order importance for iron supply from the interior to the surface*'. The above is not to suggest that the authors are unfairly duplicating finding, but rather it is meant to suggest that the section on eddy parametrizations might be the most important new result of this particular manuscript...

We thank the reviewer for reading our JAMES paper. We absolutely acknowledge that these two studies are based on the same numerical simulations. However, they are very distinct in overall goals and intended audience. For this reason, we did not inform NCOMMs regarding the other manuscript at the time of submission--we felt that it did not meet the definition of a "closely related manuscript." This is clearly a subjective question, and we apologize if we misinterpreted the policy. In any case, as we explain below, and as the reviewer appears to agree, these papers are quite distinct, and we feel strongly that the central results of this manuscript, which are not reported at all in the other paper, will be of great interest to the NCOMMs readership. While as the JAMES paper examines the dynamical mechanisms of eddy iron transport itself, this present manuscript focuses on the biogeochemical effects of this transported iron on primary productivity in the open Southern Ocean, including many novel discoveries not reported elsewhere.

In response to this issue, we have edited the abstract (lines 15-19), introduction (lines 38, 56-57), main body (lines 69-74) and conclusions (lines 241-242, 245-248), to emphasize the differences from the JAMES paper, particularly the overall ecosystem response to resolution, the year-round influence of eddies, and the potential to capture the iron fluxes via eddy parametrization.

Minor points

l40: Generally the submesoscale is dynamically defined as having both Rossby and Richardson number of order unity.

We have added the '*Richardson number*'.

l163: sp. coarser

Thank you for pointing out this typo.

I172: You currently give short explanations of the physics captured by each parametrization in the bulleted list, but I think it would help the reader if you brought the short physical descriptions out to where you first introduce the 3 runs you perform.

We have adopted the reviewer's suggestion and added the descriptions where we first introduce the three (GM, Redi and MLI) parametrizations (lines 182-185).

Reviewer #2

Minor points

I107-109 - An interesting statement but what does it imply? MLI is not important? Make some statement otherwise it is not clear why this sentence is needed.

We have added the sentence: "This suggests that energetic vertical velocities alone are not a sufficient proxy for vertical tracer transport but need to correlate with tracer concentration" (lines 119-120).

I188-192 - I suggest you remove the KPP term from the simulated iron flux in the 100km RM+R run to make it easier to compare to the 2km simulation that permits sub-mesoscale stirring. I think this change along with your statement the KPP mixing in the 100km GM+R is confined to the ML (consistent with the 2km run) would be sufficient.

Although we understand the reviewer's point, this particular configuration of parameterizations in MITgcm does not allow one to separate the GM advection and Redi diffusion from the KPP flux in output diagnostics. We have clarified the text at line 207 to ensure that the figures are easily comprehended.

I225-226 - Any thoughts on how much more Fe supply could increase if one resolved sub mesoscale processes?

Until the turbulence field numerically converges, our results (Fig. 4) suggest that eddy iron supply would continue to increase with model resolution. However, it is difficult to anticipate how far the 2km simulation may be from convergence. As the resolution is increased, processes that KPP parametrizes (e.g. convective plumes, langmuir cells and symmetric instability etc.) will also start to get resolved, and their role in transporting tracers across the ML base is still a topic of intense research.

REVIEWERS' COMMENTS:

Reviewer #1 (Remarks to the Author):

The author has satisfactorily addressed my prior concerns and I recommend the manuscript for publication.

Reviewer #2 (Remarks to the Author):

The revised paper has address my minor points and I recommend publishing the paper.